# Recurrent Endometrial Cancer: Which Is the Best Treatment? Systematic Review of the Literature

**DOI:** 10.3390/cancers14174176

**Published:** 2022-08-29

**Authors:** Stefano Restaino, Giorgia Dinoi, Eleonora La Fera, Benedetta Gui, Serena Cappuccio, Maura Campitelli, Giuseppe Vizzielli, Giovanni Scambia, Francesco Fanfani

**Affiliations:** 1Division of Obstetrics and Gynecology, University Hospital of Udine, Azienda Sanitaria Universitaria Friuli Centrale, 33100 Udine, Italy; 2Dipartimento della Salute della Donna, del Bambino e di Sanità Pubblica, Fondazione Policlinico Universitario A. Gemelli IRCCS, 00168 Rome, Italy; 3Dipartimento Scienze della Vita e Sanità Pubblica, Università Cattolica del Sacro Cuore, 00168 Rome, Italy; 4Area Diagnostica per Immagini, Dipartimento Diagnostica per Immagini, Radioterapia Oncologica ed Ematologia, Fondazione Policlinico Universitario A. Gemelli IRCCS, 00168 Rome, Italy; 5Fondazione Policlinico Universitario A. Gemelli IRCCS, UOC di Radioterapia, Dipartimento di Scienze Radiologiche, Radioterapiche ed Ematologiche, 00168 Roma, Italy; 6Department of Medicinal Area (DAME) Clinic of Obstetrics and Gynecology, Santa Maria della Misericordia Hospital, Azienda Sanitaria Universitaria Friuli Centrale, 33100 Udine, Italy

**Keywords:** recurrent endometrial cancer, secondary cytoreductive surgery, chemotherapy, radiotherapy

## Abstract

**Simple Summary:**

Endometrial cancer is the most common gynaecological tumour in developed countries. The aim of this systematic review is to compare different therapeutic strategies in the treatment of endometrial cancer recurrence to evaluate their prognostic and curative effects. The treatment of choice should be assessed according to the relapse location and to the presence of single or multiple lesions. A crucial role is also played by the type of adjuvant treatment received at the time of the first diagnosis. The molecular pattern will also be investigated in future studies. This will make it possible to personalise treatments.

**Abstract:**

Background: Endometrial cancer is the most common gynaecological tumour in developed countries. The overall rate of relapse has remained unchanged in recent decades. Recurrences occur in approximately 20% of endometrioid and 50% of non-endometrioid cases. The aim of this systematic review is to compare different therapeutic strategies in the treatment of endometrial cancer recurrence to evaluate their prognostic and curative effects based on site and type of recurrence. Methods: This systematic review of literature was conducted in accordance with the PRISMA guidelines. The study protocol was registered on PROSPERO (CRD42020154042). PubMed, Embase, Chocrane and Cinahl databases were searched from January 1995 to September 2021. Five retrospective studies were selected. Results: A total of 3571 studies were included in the initial search. Applying the screening criteria, 299 articles were considered eligible for full-text reading, of which, after applying the exclusion criteria, 4 studies were selected for the final analysis and included in the systematic review. No studies were included for a quantitative analysis. We divided the results according to the location of the recurrence: locoregional recurrence, abdominal recurrence and extra abdominal recurrence. Conclusion: the treatment of choice should be assessed according to the relapse location and to the presence of single or multiple lesions. A crucial role in the decision-making algorithm is also the type of adjuvant treatment received at the time of the first diagnosis.

## 1. Introduction

Endometrial cancer (EC) is the most common gynaecological tumour in developed countries [1]. The prevalence of EC continues to increase alongside the increasing prevalence of its risk factors including obesity and metabolic syndrome, and as the result of a growing aged population [2]. In Italy it is the fifth most frequent tumour in the female (4.6%) and the third in the group of women aged 50–69 (7%) with 8335 new cases in the 2020 [3]. Given its symptomatic nature, EC is mostly diagnosed at an early stage. 

The prognosis in these patients is favourable, with a five-year overall survival rate from 74% to 91% [4]. The overall relapse rate has remained unchanged in recent decades. Recurrences occur in approximately 20% of endometrioid (i.e., type-I histology) and 50% of non-endometrioid cases (i.e., type-II histology) [5]. The risk of developing recurrence is associated with stage, grading, tumour size, lymphovascular-space invasion (LVSI), depth of myometrial invasion, and histotype [6,7,8]. 

Barlin et al. described that in most patients who died from a recurrence of EC there were abdominal or distant diseases [9]. There is no agreement and no published prospective studies or randomized clinical trials on the type of care options in patients with endometrial cancer recurrence. Lack of data is partially explained by the different patterns of recurrence, including disease localized to the vagina, limited to the pelvis, or as metastatic disease involving the abdominal cavity or extra abdominal disease [6]. This is also because the groups of patients are heterogeneous. Perhaps the ongoing trial will help us to provide some answers [10]. 

A more detailed understanding of the prognoses associated with recurrences of endometrial cancer is necessary to improve primary as well as secondary treatment. The aim of this systematic review is to compare different therapeutic strategies in the treatment of endometrial cancer recurrence to evaluate their prognostic and curative effects based on site and type of recurrence. 

## 2. Materials and Methods

### 2.1. Search Strategy

A comprehensive search of the literature from January 1995 to September 2021, English language, was conducted. A systematic literature review was performed using electronic database (Ovid MEDLINE(R) and Epub Ahead of Print, In-Process and Other Non-Indexed Citations, and Daily, Ovid EMBASE, Ovid Cochrane Central Register of Controlled Trials, Ovid Cochrane Database of Systematic Reviews, and Scopus). The following key words were used in the search: surgery vs chemotherapy or radiotherapy or hormone therapy for recurrent endometrial cancer. The term “AND” was used to find the intersection. This review was conducted in accordance with the PRISMA guidelines [11,12]. Before data extraction, the review was registered with the International Prospective Register of Systematic Reviews PROSPERO (Registration No: CRD42020154042). To extend the search, related articles available in the database were examined, and all abstracts and citations were checked.

### 2.2. Data Extraction

The data were extracted by two researchers (S.R. and G.D.) independently for each eligible study comparing surgery, chemotherapy, radiotherapy and/or hormone therapy combined or alone in patients with recurrent endometrial cancer. Disagreements were solved by a third reviewer (E.L.). Full copies of these articles were obtained and the reviewers themselves have made independent extracts of the relevant data relating to the study characteristics. Studies were selected according to the criteria based on the following three items: (1) comparability of therapeutic strategies in the management of patients with endometrial recurrences; (2) comparability of the cohorts; (3) the type of outcome assessment of interest and appropriateness of follow-up. 

The primary outcome was to measure overall survival (OS) and disease-free survival (DFS) according to the different types of treatment. Disease-free survival (DFS) and overall survival (OS) were defined as the time, in months, passed from the date of diagnosis to the date of recurrence and death or last follow-up, respectively. Moreover, we aimed to show if there are differences on the patterns of relapse. The secondary outcome was to evaluate the rate of complications with the different treatments.

### 2.3. Inclusion Criteria

All the selected studies in this systematic review adhered to the following inclusion criteria: -comparison of outcomes of different therapeutic strategies in the management of patients with endometrial cancer recurrences. In particular, women treated surgically compared to those who underwent a non-surgical treatment (to radiotherapy (RT), chemotherapy, hormonal treatment and/or radio-chemotherapy);-available data on OS and DFS by clinical or surgical stage or both;-patients’ medical data;-only full-text articles were considered eligible for inclusion.

### 2.4. Exclusion Criteria

The exclusion criteria for this systematic review were as follows: -review, letters, editorials, case reports;-studies not published in English;-studies reporting on only one treatment of the EC recurrence without comparison-studies with missing data on outcomes.

## 3. Results

A total of 3571 studies were included in the initial search. 34 records were excluded because of duplicates; 3238 were excluded based on the title and abstract screening. Applying the screening criteria, 299 articles were considered eligible for full-text reading, of which, after applying the exclusion criteria, 4 studies were selected for the final analysis and included in the systematic review (Figure 1). 

Table 1 summarizes the characteristics of the included studies. They were only retrospective studies [13,14,15,16]. No studies were included for a quantitative analysis. For narrative purposes we divided the results according to the location of the recurrence:

-Locoregional recurrence: vaginal only or pelvic (which could also have concurrent vaginal recurrence)Abdominal recurrence: Greater pelvis (pelvic sidewall, pelvic or paraortic lymph nodes, and sigmoid colon) Abdomen (surface of liver, omentum and abdominal wall)Extra abdominal recurrence: distance recurrence out of abdomen

### 3.1. Locoregional Recurrence

In 2015, Hardarson et al. conducted a retrospective cohort study with the aim of evaluating differences in the outcome of radiotherapy vs. surgical treatment in patients with vaginal vault recurrence [14]. They identify 33 patients with recurrent endometrial cancer on vaginal vault; none of these have received adjuvant treatment. All the patients were managed with curative intent. In detail, 26 received radiotherapy, 5 received surgical treatment and 2 received a combination of the two treatment modalities. In the surgical group all patients did not develop a recurrence or death during the two years of follow-up; 40% of patients treated with radiotherapy recurred during the two years of follow-up. Two years’ survival rate was 83% for patients who received radiotherapy, and 100% for those who underwent surgery. Univariate analysis of risk factors was calculated for both re-recurrence and survival rate and any factor was statistically significant (age, FIGO stage, grade, myometrial invasion and tumour size).

Francis et al. presented a multicentre study to examine the treatment factors associated with improved outcomes in patients with endometrial cancer recurrence [13]. The authors describe a median follow up of 6.1 years, 194 women experienced a recurrence, 99 (3.7%) had locoregional recurrence, of these 43 on vaginal vault, 56 pelvic, and 95 distant recurrences. Of those patients 108 (56%) received adjuvant radiotherapy with 12.9% experiencing re-recurrence, 86 (44%) did not receive any adjuvant therapy with 53% experiencing re-recurrence. The proportion of patients with LRR who underwent radiotherapy was (72%), surgery (35%), chemotherapy (CT) (31%), hormonal therapy (9%), and 6% did not receive any salvage treatments. 40 patients (40.4%) were saved and did not experience a second failure after the initial LRR. Instead, 49 women (49.5%) with initial LRR went on to distant failure and 10 women (10.1%) had a repeat LRR. The median OS was of 14.0 years, 1.2 years and 1.0 years in women with only vaginal, pelvic and distance recurrence respectively. In multivariate analysis: salvage radiotherapy was the only factor associated with improved OS (HR 0.1 *p* = 0.04), in the case of vaginal-only recurrences; salvage surgery, salvage radiotherapy and salvage CT were associated with improved OS (HR 0.3; HR 0.3 and HR 0.1 with *p*-value = 0.04; 0.01 and <0.01 respectively) in patients with pelvic recurrences. The authors described how, in the case of pelvic recurrences, CT increases the OS, and finally a multimodality treatment could be indicated with pelvic recurrences. 

### 3.2. Abdominal Recurrence

In 2019 McAlarnen et al. conducted a retrospective study to analyse the survival and toxicity outcomes of patients with pelvic and abdominal recurrences (vaginal recurrences were excluded) of endometrial cancer [16]. There were 22 patients, 13 with pelvic recurrence and 9 with abdomen recurrence. Of these patients, 13 (59.1%) were included in the multimodality cohort, 4 (18.2%) in the surgery cohort, and 5 (22.7%) in the non-surgery cohort (CT or EBRT). The multimodality cohort included patients who underwent a combination of surgery, EBRT, and CT with or without BRT. Two years’ DFS was greater in the multimodality cohort than in the non-surgery and surgery cohorts. Similar results were obtained in terms of OS, where it was greater in patients who had received multimodality therapy (68%) and surgery (67%); however, it was lower in patients of the non-surgery group (53%). At 40 months of follow up, only patients treated with multimodality cohort were still alive. The authors concluded that multimodality therapy is feasible and well tolerated in selected patients with isolated recurrences to the pelvis and peritoneal cavity.

### 3.3. Extra-Abdominal Recurrence

At least for extra-abdominal recurrence, Dowdy identifies 82 patients, 28 with isolated lung recurrence and 54 with multiple sites of recurrence [15]. Of those with isolated lung recurrence, 19 (68%) patients were treated with hormonal therapy, 9 (32%) with CT and 14 (50%) with surgical treatment. Disease-specific survival was 50% (27 months’ median survival) in patients with isolated lung recurrence, compared to 4% in patients with multiple sites of recurrence (7 months’ median survival). Through the univariate analysis similarly multiple parameters have been found predictive for OS after the recurrence (tumour grade, max diameter of recurrence and treatment with CT). Patients with a grade 3 tumour had worse OS compared with grade 1 or 2 (*p* = 0.01). In the case of pulmonary nodules less than 2 cm the authors describe a 29 months’ median OS compared to 13 months for those patients with larger pulmonary nodules (*p* = 0.005). 

The authors observed that the different treatment (hormonal therapy, surgery or combination therapy) was not predictive of survival after recurrence. Median OS was 14 months for those who received CT as part of their treatment for recurrence, compared to 28 months for those who did any other treatment (surgery or hormonal with *p* = 0.04). On the other hand, median OS was 28 months for patients treated with hormonal therapy only, compared to 18 months for those treated only surgically (*p* = NS). 

## 4. Discussion

The recurrence rate in patients with early endometrial cancer is about 10–15% [1,2]. This value increases according to the stage and the histotype. Given the relatively low numbers compared to the incidence rate, it is understandable that we do not have clear data on the appropriate treatment of the recurrence. Furthermore, in literature there are few prospective studies or trials focused on the treatment of recurrent endometrial cancer, and even fewer are studies that give an overview of treatment choices based on different sites of recurrence [17]. Despite several studies having identified different patterns of relapse [18,19], we still do not have unanimous consensus about the different strategies that we could apply for different relapse localizations. The treatment of recurrent endometrial cancer depends primarily on prior therapy, and subsequently on the site of recurrence, and the patient’s performance status [17]. Some studies have examined the correlation between clinical patterns of relapse and the clinic-pathologic features of primary disease [20,21]. Furthermore, data on clinical outcomes of recurrent endometrial carcinoma in terms of disease type and targeted treatment are not conclusive. [22,23,24,25]. Several studies have shown how the presence of single or multiple relapses influences OS and DFS [25,26,27,28]. In this context, patients with locoregional recurrence only, long-term remissions may be achieved. Instead, for those in whom endometrial cancer recurs or progresses in distant sites, the treatment is often palliative than curative [16,29]. 

Surgery represents a cornerstone of initial diagnosis of endometrial cancer but no more data are available for surgical treatment in advanced and recurrent endometrial cancer. Typically, a combination of surgery, radiotherapy and/or CT is employed. A multidisciplinary approach is essential to define the more appropriate treatment [30]. Current recommendations are based on consensus guidelines and retrospective data [6,20,21]. Several studies have tried to assess the effectiveness of each individual treatment on recurrence [18], but few have performed a comparison between different treatment options. The same guidelines leave a wide margin of choice on the type of treatment to be offered to these patients. For example, the international guidelines NCCN divide on localized and disseminate disease [21]. Instead, at the ESMO-ESGO- ESTRO Consensus conference on endometrial cancer they talk about the optimal systemic therapies for advanced/recurrent disease [6,30]. For this reason, we have tried to divide the recurrence according to the site of onset and assess which may be the most appropriate treatment (Figure 1).

### 4.1. Locoregional Recurrence

Patients with locoregional recurrences have a better prognosis in the case of isolated vaginal vault disease compared with relapses in another pelvic site [31]. The choice of treatment of vaginal recurrence should take into account previous RT and DFS, as reported in the literature [31]. The standard approach of vaginal recurrence could be radiation therapy (EBRT plus BRT) in patients who have never undergone radiotherapy treatment [30]. In the setting of prior pelvic radiation, treatment recommendations are even less well defined, and the optimal treatment remains unclear. In PORTEC-1 trial five-years’ survival rates after vaginal recurrence was 65% in the non-irradiate group versus 43% in the irradiate group [5]. In conclusion, radiotherapy is typically the treatment of choice for a local relapse occurring in an RT-naive area.

The role of surgery is debated in recurrent endometrial cancer. The study of Hardarson describes a radiotherapy-naïve population. The results indicate that surgical removal of the recurrence is an effective treatment [14]. For these reasons, in well-selected patients, radical surgery could be considered with the aim to achieve negative margins [28]. Historically, surgery has focused on total pelvic exenteration as a potential treatment for advanced recurrent disease in the pelvis [32,33,34]. This surgery has a high rate of morbidity (41% of major complications) and peri-operative mortality (4.8%) rates [33]. If surgery is not feasible, an option includes stereotactic body radiotherapy targeting the recurrence, permanent seed implants, or proton therapy. In selected cases, limited volume re-irradiation with EBRT and BRT may be an option in patients who only had previous BRT [6,30]. The best results occur when the recurrence interval is long and the recurrence measures under 4 cm [30]. Francis et al. found that salvage radiotherapy was the only factor associated with improved OS in patients with vaginal vault recurrence [13].

Current recommendations are based on consensus guidelines and retrospective data. We have only one ongoing study GOG 238 [10]. This randomized phase II trial studies radiation therapy and cisplatin to see how well they work compared with radiation therapy alone in recurrent carcinoma limited to the pelvis and vagina. 

As summarised in Figure 2, we could affirm that in the case of:

-central pelvic recurrence; the treatment of choice is surgery or radiation therapy based according to previous RT, size of disease, complete removal of macroscopic disease, easily accessible vaginal tumour. -regional pelvic recurrences; the treatment is radiation therapy, associated if possible with chemotherapy, or we can also consider exenterative surgery in selected patients.

### 4.2. Abdominal Recurrence

Multimodality approach, including a combination of surgery, radiotherapy and/or chemotherapy should be considered for patients with abdominal recurrence. These patients should be treated in referral centers, because they require a multidisciplinary approach and highly surgical skills. Scarabelli in 1990 was the first to investigate the role of cytoreductive surgery in recurrent endometrial cancer with a small series of patients [35]. The same results were reported twenty years later by different authors [19,36,37]. The authors have demonstrated that the residual tumour at the end of surgery was the only significant variable involved in PFS and OS [19,35,36,37]. This kind of surgery results on perioperative morbidity and mortality; for this reason, it is necessary to treat a careful selection of patients. Moreover, additional therapy systems are necessary for the completion of the therapeutic program. In the article of McAlanern, despite salvage treatment was being statistically significant, the patient with multimodality treatment experienced a lower rate of recurrence [16]. 

Palliative surgery should be considered to control symptoms (bleeding fistula or bowel occlusion) [30].

### 4.3. Extra-Abdominal Recurrence

Hematogenous recurrence in endometrial cancer can involve brain, lungs, liver and bone. The most common site of hematogenous relapse is pulmonary with an incidence of 2.3% to 7% [38,39,40,41,42,43]. We can consider different options of treatment for patients with disseminate metastases or patients with isolated extra abdominal metastases. We have to consider that endometrial cancer recurring after first line chemotherapy is frequently a chemo-resistant disease. The main predictors of response in the treatment of metastatic disease are well differentiated tumours, a long DFS, the location and extent of extrapelvic (particularly pulmonary) metastases [39,40,41,42,43,44]. In distant isolated recurrences, surgical resection is feasible in selected patients with limited and resectable disease [28]. In 2004, Anraku found that pulmonary metastasectomy for uterine malignancies is a safe and acceptable treatment to improve survival with 1% of morbidity and mortality [44]. In patients with disseminate metastases, asymptomatic, low-grade with hormonal receptor, one option for treatment includes hormonal therapy followed by systematic therapy on progression [30]. Symptomatic, high-grade large-volume metastasis can be treated with systemic therapy with or without RT [21]. 

Not least, we have to consider progressive development in the field of molecular biology and the progress in recent years of target therapies in patients affected by endometrial cancer. The targeted therapy aims to inhibit specific molecular pathways. For example the mammalian target of rapamycin (mTOR), the angiogenesis, and the epidermal growth factor receptor (EGFR) family are relevant therapeutic targets [45,46,47,48]. 

## 5. Conclusions

In conclusion, what emerges from the current literature is the importance of choosing the type of treatment based on the location of the lesions and the presence of single or multiple lesions. A crucial role is also played by the type of adjuvant treatment received at the time of the first diagnosis. This indicates that these patients are a heterogeneous group. The molecular pattern will also be investigated in future studies. This will make it possible to personalise treatments. 

In locoregional recurrences a fundamental role is occupied by the adjuvant therapy, usually for patients who are RT naïve, the treatment of choice is RT. Although an important role is represented by the size of the tumour; in selected patients the surgical approach can be considered with or without adjuvant therapy.

In the case of abdominal recurrence, multimodality and personalised treatment should be considered. 

In patients with abdominal and extrabdominal recurrence, complete cytoreduction, when achieved, can significantly improve survival. As an alternative, hormonal treatment and clinical trial should be encouraged.

Patients with recurrent disease represent a heterogeneous group with different characteristics so prospective studies will be necessary to compare the different types of treatment according to the location of the relapse and clinical trial should be supported. Promising results are expressed by the target therapy. 

## Figures and Tables

**Figure 1 cancers-14-04176-f001:**
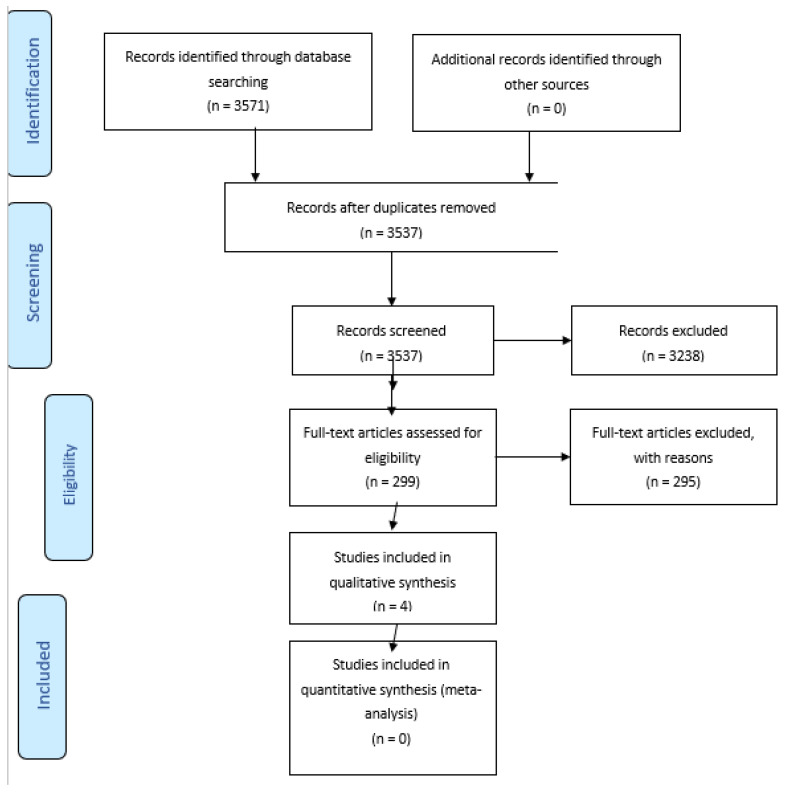
PRISMA 2009 Flow Diagram.

**Figure 2 cancers-14-04176-f002:**
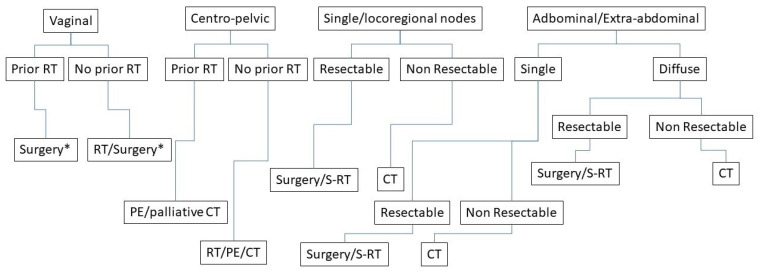
Flowchart of treatment of recurrence. RT = radiotherapy; CT = chemotherapy; * Partial/Total colpectomy.

**Table 1 cancers-14-04176-t001:** Characteristics of the studies.

Author, Year	Type of Study	Sample (N)	Site of First Recurrence (N)	Type of Treatment (N)		Secondary Recurrence Rate (%)		2-Year Desease Free Survival (%)		Overall Survival (Median, Years)
Hardarson HA, 2015	Retrospective	33	Vaginal, 33	RT Surgery RT + Surgery	26 (78.8) 5 (15.1) 2 (6.1%)	RT Surgery RT + Surgery	40% 0% 0%	RT Surgery RT + Surgery	83% 100% 100%	Not Analysed
Francis SR, 2019	Retrospective	194	Vaginal, 43	RT Surgery CHT Hormonal Combined tx None	24 (55.8%) 3 (7%) 0 (0%) 0 (0%) 16 (37.2%) 0 (0%)	* LLR NONE DISTANCE	10 (10.1%) 40 (40.4%) 49 (49.5%)	** LLR DISTANCE NONE	50% 14.3% 94.1%	14
			Pelvic, 56	RT Surgery CHT Hormonal Combined tx None	9 (16%) 6 (10.7%) 2 (3.6%) 2 (3.6%) 31 (55.4%) 6 (10.7%)					1.2
			Distance, 96	Not Analysed		Not Analysed		Not Analysed		1.0
Mc Alarnen L, 2019	Case series Retrospective	22	Pelvic, 13	Surgery No Surgery Combined treat.	2 (15.4%) 3 (23.1%) 8 (61.5%)	Surgery No Surgery Combined treat.	1 (50%) 1 (33.3%) 2 (24%)	Not Analysed		Not Analysed
			Abdominal, 9	Surgery No Surgery Combined treat.	2 (22.2%) 2 (22.2%) 5 (55.6%)	Surgery No Surgery Combined treat.	1 (50%) 2 (100%) 2 (40%)	Not Analysed		
			Pelvic + Abdominal, 22	Surgery No Surgery Combined treat.	4 (18%) 5 (22.7%) 13 (59.1%)	Surgery No Surgery Combined treat.	2 (40%) 3 (60%) 4 (31%)	Surgery No Surgery Combined treat.	67% 53% 68%	
Dowdy SC, 2007	Retrospective	82	Pulmonary Isolated, 28	CHT Hormonal Surgery Combined treat.	9 (32%) 19 (68%) 14 (50%) 11 (13%)	Not Analysed			50%	
			Multiple site, 54 (Abdominal and extrabdominal)						4%	

* the authors analysed vaginal and pelvic together as ‘locoregional secondary recurrence’. ** the authors analysed vaginal and pelvic together as” locoregional survival rate”. CHT chemotherapy, LRR locoregional recurrence, RT radiotherapy.

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
