# Peer review of "Recurrent Endometrial Cancer: Which Is the Best Treatment? Systematic Review of the Literature"

_cancers, 2022, doi:10.3390/cancers14174176_

Round 1
Reviewer 1 Report
This is a well written article. I enjoyed reading it but it contains nothing about specific systemic treatment.
What is the role of chemotherapy what is the role of immuncheckpoint inhibition. All of these are new but standard treatment. If the article is only going to address RT and surgery say so. They do mention the import of the type of prir treatment. they then need to address this issue
Author Response
This is a well written article. I enjoyed reading it but it contains nothing about specific systemic treatment.
What is the role of chemotherapy what is the role of immuncheckpoint inhibition. All of these are new but standard treatment. If the article is only going to address RT and surgery say so. They do mention the import of the type of prir treatment. they then need to address this issue
I thank the reviewers for their comments. The aim of this meta-analysis was to find articles that could compare different treatment strategies. In fact, we tried to compare different therapeutic strategies in the treatment of endometrial cancer recurrence in order to evaluate their prognostic and curative effects based on site and type of recurrence. We agreed with the reviewer's comments and added a specification on systemic and target treatment in “discussion” section.
Reviewer 2 Report
Overall a good review manuscript of "Recurrent Endometrial Cancer: Which is the beast Treatment? Systemic Review of the Literature."
The only reservation is regarding the small selection of studies that were appropriate for the final analysis and which were Included in the systemic review. Otherwise the manuscript was well written with adequate referencing provided.
There are minor grammatical errors which I have listed below for your consideration:
1. Line 131 - "Extra abdominal recurrence: distance recurrence out of abdominal." Could this read Extra abdominal recurrence: distance recurrence out of abdomen.
2. Line 255 - "In this contest, patients with locoregional recurrence only, long-term remissions may be achieved." Could this read - In this context, patients with locoregional recurrence only, long-term remissions may be achieved.
3. Line 267 - "Instead, on ESMO-ESGO- ESTRO Consensus conference on endometrial cancer they talk about the optimal systemic therapies for advanced/ recurrent disease [6,30]." Could the sentence read - Instead, at the ESMO-ESGO- ESTRO Consensus conference on endometrial cancer they talk about the optimal systemic therapies for advanced/ recurrent disease [6,30].
4. Line 273 - "The choose of treatment of vaginal recurrence should take into account previous RT and the DFS, as reported in literature [31]." Could the sentence read - The choice of treatment of vaginal recurrence should take into account previous RT and the DFS, as reported in the literature [31].
5. Line 278 - "In PORTEC-1 trial 5-years survival rate after vaginal recurrence were 65% in the non-irradiate group versus 43 % in the irradiate group [5]." Could the sentence read - In PORTEC-1 trial 5-years survival rates after a vaginal recurrence was 65% in the non-irradiated group versus 43 % in the irradiated group [5].
6. Line 324 - "We can consider different option of treatment for patients with disseminate metastases or patient with isolate extra abdominal metastases. We have to consider that endometrial cancer recurring after first line chemotherapy is frequently a chemo-resistant disease." Could the line read - We can consider different options of treatment for patients with disseminate metastases or patients with isolated extra abdominal metastases. We have to consider that endometrial cancer recurring after first line chemotherapy is frequently a chemo-resistant disease.
6. Line 330 -"Anraku find that pulmonary metastasectomy for uterine malignancies is a safe and acceptable treatment to improve survival with 1% of morbidity and mortality [44]." Could the sentence read Anraku found that pulmonary metastasectomy for uterine malignancies is a safe and acceptable treatment to improve survival with 1% of morbidity and mortality [44].
7. Line 349 - "In patients with abdominal and extrabdominal recurrence, omplete cytoreduction when achieved, can significantly improve survival." could the sentence read -In patients with abdominal and extrabdominal recurrence, complete cytoreduction when achieved, can significantly improve survival.
Author Response
Overall a good review manuscript of "Recurrent Endometrial Cancer: Which is the beast Treatment? Systemic Review of the Literature."
The only reservation is regarding the small selection of studies that were appropriate for the final analysis and which were Included in the systemic review. Otherwise the manuscript was well written with adequate referencing provided.
We thank the reviewers for their comments. The small number of articles included is due to the fact that we looked for work comparing different treatments.
There are minor grammatical errors which I have listed below for your consideration:
- Line 131 - "Extra abdominal recurrence: distance recurrence out of abdominal." Could this read Extra abdominal recurrence: distance recurrence out of abdomen.
We have accepted the reviewer's suggestion and modified it.
- Line 255 - "In this contest, patients with locoregional recurrence only, long-term remissions may be achieved." Could this read - In this context, patients with locoregional recurrence only, long-term remissions may be achieved.
We thank the reviewers. We have modified it.
- Line 267 - "Instead, on ESMO-ESGO- ESTRO Consensus conference on endometrial cancer they talk about the optimal systemic therapies for advanced/ recurrent disease [6,30]." Could the sentence read - Instead, at the ESMO-ESGO- ESTRO Consensus conference on endometrial cancer they talk about the optimal systemic therapies for advanced/ recurrent disease [6,30].
We have accepted the reviewer's suggestion and modified it.
- Line 273 - "The choose of treatment of vaginal recurrence should take into account previous RT and the DFS, as reported in literature [31]." Could the sentence read - The choice of treatment of vaginal recurrence should take into account previous RT and the DFS, as reported in the literature [31].
We collected the reviewers suggestion. We have modified it.
- Line 278 - "In PORTEC-1 trial 5-years survival rate after vaginal recurrence were 65% in the non-irradiate group versus 43 % in the irradiate group [5]." Could the sentence read - In PORTEC-1 trial 5-years survival rates after a vaginal recurrence was 65% in the non-irradiated group versus 43 % in the irradiated group [5].
We thank the reviewers. We have modified it.
- Line 324 - "We can consider different option of treatment for patients with disseminate metastases or patient with isolate extra abdominal metastases. We have to consider that endometrial cancer recurring after first line chemotherapy is frequently a chemo-resistant disease." Could the line read - We can consider different options of treatment for patients with disseminate metastases or patients with isolated extra abdominal metastases. We have to consider that endometrial cancer recurring after first line chemotherapy is frequently a chemo-resistant disease.
We collected the reviewers suggestion. We have modified it.
- Line 330 -"Anraku find that pulmonary metastasectomy for uterine malignancies is a safe and acceptable treatment to improve survival with 1% of morbidity and mortality [44]." Could the sentence read Anraku found that pulmonary metastasectomy for uterine malignancies is a safe and acceptable treatment to improve survival with 1% of morbidity and mortality [44].
We have accepted the reviewer's suggestion and modified it.
- Line 349 - "In patients with abdominal and extrabdominal recurrence, omplete cytoreduction when achieved, can significantly improve survival." could the sentence read -In patients with abdominal and extrabdominal recurrence, complete cytoreduction when achieved, can significantly improve survival.
We thank the reviewers. We have modified it.
Reviewer 3 Report
The authors performed a systematic review summarizing the evidence on recurrent endometrial cancer. While the review is nicely performed, its findings are similar to those of other reviews in the field
Am J Clin Oncol. 2015 Apr;38(2):206-12. doi: 10.1097/COC.0b013e31829a2974.
Expert Rev Anticancer Ther. 2018 Sep;18(9):873-885. doi: 10.1080/14737140.2018.1491311.
; hence, the authors should try to comment the novelty of their study.
Furthermore i am quite interested to know why they chose to omit studies involved in targeted therapies
Curr Oncol Rep. 2021 Nov 4;23(12):139. doi: 10.1007/s11912-021-01129-4.
Author Response
The authors performed a systematic review summarizing the evidence on recurrent endometrial cancer. While the review is nicely performed, its findings are similar to those of other reviews in the field
Am J Clin Oncol. 2015 Apr;38(2):206-12. doi: 10.1097/COC.0b013e31829a2974.
Expert Rev Anticancer Ther. 2018 Sep;18(9):873-885. doi: 10.1080/14737140.2018.1491311.
; hence, the authors should try to comment the novelty of their study.
Furthermore i am quite interested to know why they chose to omit studies involved in targeted therapies
Curr Oncol Rep. 2021 Nov 4;23(12):139. doi: 10.1007/s11912-021-01129-4.
I thank the reviewers for their comments. The aim of this meta-analysis was to find articles that could compare different treatment strategies. In fact, we tried to compare different therapeutic strategies in the treatment of endometrial cancer recurrence in order to evaluate their prognostic and curative effects based on site and type of recurrence. We agreed with the reviewer's comments and added a specification on systemic and target treatment in “discussion” section.
Round 2
Reviewer 1 Report
I feel the the authors have revised appropriately
Reviewer 3 Report
This article can now be accepted with major text editing